

 

# Channel flow, tectonic overpressure, and exhumation of
# high-pressure rocks in the Greater Himalayas
**Fernando O. Marques[1*], Nibir Mandal[2], Subhajit Ghosh[3], Giorgio Ranalli[4],**
**Santanu Bose[3]**
*[1]Universidade de Lisboa, Lisboa, Portugal*
*[2]Jadavpur University, Kolkata, India*
*[3]University of Calcutta, Kolkata, India*
*[4]Carleton University, Ottawa, Canada*
**Abstract**
The Himalayas are the archetype of continental collision, where a number of long-
standing fundamental problems persist in the Greater Himalayan Sequence (GHS): (1)
contemporaneous reverse and normal faulting; (2) inversion of metamorphic grade; (3) origin of
high- (HP) and ultra-high (UHP) pressure rocks; (4) mode of ductile extrusion and exhumation
of HP and UHP rocks close to the GHS hanging wall; (5) flow kinematics in the subduction
channel; and (6) tectonic overpressure, here defined as $TOP = P/P_L$ where $P$ is total (dynamic)
pressure and $P_L$ is lithostatic pressure. In this study we couple Himalayan geodynamics to
numerical simulations to show how one single model, upward-tapering channel (UTC) flow, can
be used to find a unified explanation for the evidence. The UTC simulates a flat-ramp geometry
of the main underthrust faults, as proposed for many sections across the Himalayan continental
subduction. Based on the current knowledge of the Himalayan subduction channel geometry and
geological/geophysical data, the simulations predict that a UTC can be responsible for high $TOP$
(> 2). $TOP$ increases exponentially with decrease in UTC's mouth width, and with increase in
underthrusting velocity and channel viscosity. The highest overpressure occurs at depths < -60





km, which, combined with the flow configuration in the UTC, forces HP and UHP rocks to
exhume along the channel's hanging wall, as in the Himalayas. By matching the computed
velocities and pressures with geological data, we constrain the GHS's viscosity to be $\leq 10^{21}$ Pa s,
and the effective convergence (transpression) to a value $\leq 10\%$. Variations in channel dip over
time (> or < 15º) may promote or inhibit exhumation, respectively. Viscous deformable walls do
not affect overpressure significantly for a viscosity contrast (viscosity walls/viscosity channel) in
the order of 1000 or 100. *TOP* in a UTC, however, is only possible if the condition at the bottom
boundary is no outlet pressure; otherwise it behaves as a leaking boundary that cannot retain
dynamic pressure. However, the cold, thick and strong lithospheres forming the Indian and
Eurasian plates are a good argument against a leaking bottom boundary in a flat-ramp geometry,
and therefore it is possible for overpressure to reach high values in the GHS.

Keywords: Himalayan geodynamics; channel flow; Greater Himalayas; numerical modelling;
tectonic overpressure; exhumation HP and UHP rocks

*Corresponding author. E-mail address: fomarques@fc.ul.pt

**1. Introduction**

Continental collision has brought together two continents, India and Eurasia, which were

previously separated by thousands of kilometres of oceanic lithosphere that has been consumed
by subduction. Understanding the mechanics of the collisional interface, known as the Greater
Himalayas Sequence (GHS), has continuously stimulated geoscientists to search for new
concepts/models. Most critically, high- (HP) and ultrahigh- (UHP) pressure rocks crop out along
the Himalayan GHS, thus raising long-standing and lively debated questions regarding formation
and exhumation of HP and UHP rocks, and the difference between lithostatic and dynamic



pressures (overpressure) in dynamic systems. The GHS appears therefore as a unique natural
prototype that can be modelled numerically in the search for answers to those critical questions.

*Figure 1. Geological setting of the eastern Himalayas, highlighting the architecture of its major tectonic*
*elements. A – Simplified geological map of the eastern Himalayas (adapted from Grujic et al., 2011;*
*Unsworth et al., 2005). White line along 90°E marks the cross-section shown in B. B – Schematic section*
*across the Himalayas (adapted from Grujic et al., 2011), in which the UTC stands out (GHS in red). The*



*GHS is bounded at the top by the South Tibet Detachment (STD) and at the bottom by the Main Central*
*Thrust (MCT). MHT – Main Himalayan Thrust. C – Model setup of the UTC, with shape and dimensions*
*similar to the natural prototype in B. The "foot wall" (moving wall) and the "hanging wall" (no slip*
*wall) correspond to the MCT and the STD, respectively. Apart from the later folding of both MCT and*
*STD, the similarity between nature and model setup is apparent.*

*1.1. Geological setting*

Based on metamorphic grade and structural style, four units and the major faults

separating them were distinguished by Gansser (1964), which are from bottom to top (Fig. 1):
Sub-Himalayan Sequence (SHS – unmetamorphosed Tertiary rocks), Main Boundary Thrust
(MBT), Lesser Himalayan Sequence (LHS – low-grade metamorphic rocks), Main Central
Thrust (MCT), Greater Himalayan Sequence (GHS – high-grade metamorphic rocks), South
Tibetan Detachment (STD), and Tethyan Sedimentary Sequence (TSS – unmetamorphosed to
weakly metamorphosed rocks). All the main faults are N-dipping thrusts, except the STD that
also dips to N but is a normal fault.

The GHS shows patchy occurrences of eclogites close to the STD (Grujic et al., 2011;

Ganguly et al., 2000; O'Brien et al., 2001; Groppo et al., 2007; Corrie et al., 2010; Kellett et al.,
2013; Sorcar et al., 2014; Zhang et al., 2015) (Fig. 1A). Recent petrologic studies provide
estimates for spatial-temporal variations of pressure ($P$) and temperature ($T$) in the GHS. The
peak metamorphic conditions are $T \sim 760\ °C$ and $P \geq 1.5\ GPa$ for eclogitization in the Bhutan
Himalayas (Grujic et a., 2011). Peak conditions with $T = 670\ °C$ and $P \geq 1.5\ GPa$ were reported
for the Nepal Himalayas (Corrie et al., 2010). On the other hand, an estimate of the metamorphic
peak at $P = 2.7–2.9\ GPa$ and $T = 690–750\ °C$ from coesite-bearing eclogites in the western
Himalayas was provided by O'Brien et al. (2001). The eclogites have been in part overprinted by
regionally more extensive granulite facies conditions of $800\ °C$ at $\sim 1\ GPa$ (Grujic et al., 2011;
Ganguly et al., 2000; Groppo et al., 2007; Zhang et al., 2015). *PT*-time paths suggest exhumation
of these high-grade rocks under nearly isothermal decompression after peak metamorphic
conditions (Ganguly et al., 2000; Groppo et al., 2007; Sorcar et al., 2014). Using cooling rates,





the exhumation history of the high-grade rocks was interpreted as a two-stage event by Ganguly
et al. (2000), marked by exhumation at a rate of 15 mm/yr to a depth of 15 km, followed by slow
exhumation at a rate of 2 mm/yr to a depth of at least 5 km, which occurred broadly in Miocene
times (Grujic et al., 2011; Corrie et al., 2010; Kellett et al., 2013; Sorcar et al., 2014; Warren et
al., 2011; Rubatto et al., 2013).
The exhumation mechanics of GHS rocks is one of the most debated issues in the
Himalayas (and elsewhere where HP and UHP rocks outcrop), having led to a variety of tectonic
models that postulate channel flow by topographic forcing (Wobus et al., 2005; Beaumont et al.,
2001) or transpression (Grujic et al., 1996). Grujic et al. (1996) first proposed the GHS in the
Bhutan Himalayas as deep crustal ductile rocks extruded between the MCT and the STD.
Numerical models have integrated geological, tectonic, geophysical, metamorphic and
rheological data to provide possible explanations for the exhumation process. The models
postulate a channel flow of low-viscosity rocks in the middle to lower crust, driven by
topographic pressure gradient, to account for the extrusion dynamics of high-grade metamorphic
rocks in the GHS (Wobus et al., 2005; Beaumont et al., 2001). The channel flow model can also
explain the coeval reverse and normal kinematics along the MCT and STD, respectively (Fig.
1B). However, as Grujic et al. (2011) pointed out, these models cannot "*predict the exhumation*
*of lower orogenic (>50 km, i.e. >1.4 GPa) crustal material*" in their basic form. To overcome
this limitation, an alternative exhumation mechanism was proposed by Grujic et al. (2011), with
additional tectonic forcing (transpression) by the impingement of strong Indian crust into the
already weak lower crustal granulitized eclogites below southern Tibet. However, previous
models do not comprehensively address the mechanics of overpressure leading to the formation
of eclogites (Schulte-Pelkum et al., 2005), and their focused exhumation close to the STD.
Given that the current models do not fully explain the observations in the GHS, in this
study we couple eastern Himalayan geodynamics with numerical simulations to show how one

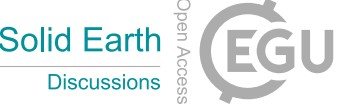

single model, upward-tapering channel (UTC) flow, as in the current eastern Himalayas (Fig.
1B), can be used to find a unified explanation for the following persisting problems: (1)
contemporaneous reverse and normal faulting; (2) inversion of metamorphic grade; (3) origin of
high- (HP) and ultrahigh- (UHP) pressure rocks; (4) mode of ductile extrusion and exhumation
of HP and UHP rocks close to the GHS hanging wall (STD); (5) flow kinematics in the
subduction channel; and (6) tectonic overpressure.

*1.2. Premises*

We model channel flow with a linear viscous fluid by the Navier-Stokes equation with

body force (gravity), therefore pressure in the channel depends on viscosity and velocity
configuration. Most critically, the velocity field depends on channel geometry and conditions
applied at the boundaries (e.g. Marques et al., 2018). Ultimately, *TOP* can only exist if the
channel walls are strong enough. Therefore, when investigating pressure in a viscous channel,
one has to take into account four fundamental issues:
(1) *Viscosity* – the viscosity term in the Navier-Stokes equation depends on a number of

parameters, all of which are incorporated in the Arrhenius term in a constitutive equation.

Therefore, the modeller has two options when investigating the effects of viscosity on

pressure: either use a full constitutive equation and test all the parameters in the Arrhenius

term, or simply and directly vary the magnitude of the viscosity. We chose the second option

in our numerical simulations, since our focus is the assessment of parameter variations on

the development of overpressure and flow configuration.

(2) *Geometry of the channel* – given that flow configuration inside the channel plays a critical

role in the pressure distribution, we tested three main shapes of the channel: parallel-sided

(parallelepiped), and upward (similarly to Marques et al., 2018) or downward tapering

channels.





(3) *Boundary conditions* – the conditions at the boundaries can either promote or inhibit *TOP*,

because they control the flow pattern and the pressure retention inside the channel.

Therefore, we tested different velocity configurations applied at the underthrusting (foot)

wall (simple or simple+pure shears), and different conditions at the boundaries like slip, no-

slip or outlet pressure.

(4) *How the walls of the pressure vessel react to internal pressure* – under particular applied

boundary conditions, the Navier-Stokes equation produces *TOP* in an upward tapering

channel that can reach values orders of magnitude greater than observed in nature; therefore

we will discuss the theoretical values in view of the current knowledge on natural HP and

UHP rocks. The discussion of channel flow is similar to discussing a pressure vessel with an

overpressured fluid inside: one has to investigate the conditions to produce overpressure

inside the vessel (the channel in the prototype), and simultaneously the strength of the vessel

walls (the lithosphere in the prototype) to support the internal pressure without failure (by

brittle or viscous yield). We will therefore discuss the strength of the channel walls in view

of the current knowledge about the Indian (footwall) and Eurasian (hanging wall)

lithospheres, especially in terms of thickness and strength.


This study builds on the conceptual work by Marques et al. (2018) on tectonic

overpressure.

Given the above premises, we investigated the conditions under which overpressured

rocks can form and be exhumed in a prototype like the Himalayas: geometry of the channel,
conditions at the boundaries, applied velocities, and viscosity. Based on the numerical
simulations and the current knowledge of the Himalayas, we discuss the theoretical values of
overpressure, the obtained exhumation velocities, the most likely viscosity of the subducted
rocks, and finally the effects of the strength of the channel walls on overpressure.




## 2. Numerical modelling

We modelled the subduction channel, as illustrated in Fig. 1C, with an incompressible
linearly viscous fluid, which has been accepted as a simple but effective approximation to the
behaviour of rocks undergoing ductile flow. The setup simulates a flat-ramp geometry of the
main underthrust faults, as shown in many cross-sections of the Himalayas, in particular the one
shown in Fig. 1B. For steady-state flow of a viscous incompressible Newtonian fluid at very low
Reynolds number, the dynamic Navier-Stokes equations reduce to the Stokes approximation,
which is the basis of the COMSOL code for computational fluid dynamics used here.

*2.1. Boundary conditions and model setup*

The boundary conditions were as follows (see Fig. 1C, and Methods in Appendix for
further details): (1) slab-parallel velocity ($U$) applied on the underthrusting (foot)wall (2 to 20
cm/yr) (Feldl and Bilham, 2006; DeMets et al., 2010), and fixed hanging wall; (2) viscosity ($\eta$)
between $10^{19}$ and $10^{22}$ Pa s (Beaumont et al., 2001; England and Houseman, 1989; Copley et al.,
2011); (3) channel dip $\alpha$ (15-30º); (4) channel mouth's width $W_m$ = 25 to 100 km, and width at
the channel's base $W_b$ = 150 or 200 km, from which we define $W_m^* = W_m/W_b$; (5) constant
density of the material in the channel (2800 kg/m$^3$). Given the viscosity contrast between
foot/hanging walls of the GHS and channel material, the channel walls were assumed
undeformable in the first simulations, except when testing the effects of non-rigid walls on
overpressure.
The metamorphic processes occur in response to the total isotropic stress, called *dynamic*
*pressure*, which is a sum of the tectonic (Stokes) and lithostatic pressures ($\rho g z$, where $\rho$ is
density, $g$ is gravitational acceleration, and $z$ is depth). We evaluate the dynamic pressure to
explain the occurrence of high-pressure rocks in the GHS, and we define an overpressure factor



(*TOP*) as the non-dimensional ratio between dynamic and lithostatic pressures (Figs. A1 and
A2). For a better understanding of overpressure in a UTC, we carried out a parametric study of
*TOP* as a function of $\eta$, $W_m$, $\alpha$, $U$, and effective convergence velocity (transpression) (see
Methods in Appendix for details). The prime focus of our investigation concerned the
simulations with $U = 5$ cm/yr, $\alpha = 20º$, $W_m = 100$ km and $W_b = 150$ km, which represent the most
common and conservative values. We then use the numerical results to constrain the viscosity,
pressure and velocity in the channel, consistent with current geological data and estimates.

**3. Model results**
*3.1. Flow patterns*

The model UTC shows two main layers, one flowing downward due to applied

underthrusting motion in the footwall, and another flowing upward and so inducing relative
normal faulting on the hanging wall (Fig. 2). Two distinct flow cells exist, one as an open circuit
in the shallow channel (< 30 km depth), and another as a closed circuit in the deeper channel.
The line of flow reversal (dashed white line in Fig. 2B) acts as an internal large-scale shear zone
with curved geometry and thrust motion. The upward flowing layer shows, at shallow depth, a
maximum velocity $\approx 0.5 \times 10^{-9}$ m/s, i.e. ~16 mm/year. The line of flow convergence separates
crustal materials of contrasting pressures, one towards the footwall with $P < 1.5$ GPa, the other
towards the hanging wall with $P > 1.5$ GPa (red curve in left hand panel in Fig. 2D), which is the
pressure at which eclogite formation is possible at -30 km. Overall, the flow pattern shows that
significantly overpressured rocks (*TOP* > 2.) can be exhumed rapidly through a narrow region
close to the hanging wall of the channel, which corresponds to the STD in the Himalaya and
where HP and UHP rocks have been found.

*3.2. Dynamic pressure and overpressure*





Model results are presented as colour maps (Fig. 2) and graphs (Fig. 3), the latter
showing the effects of several parameters on overpressure in the subduction channel.

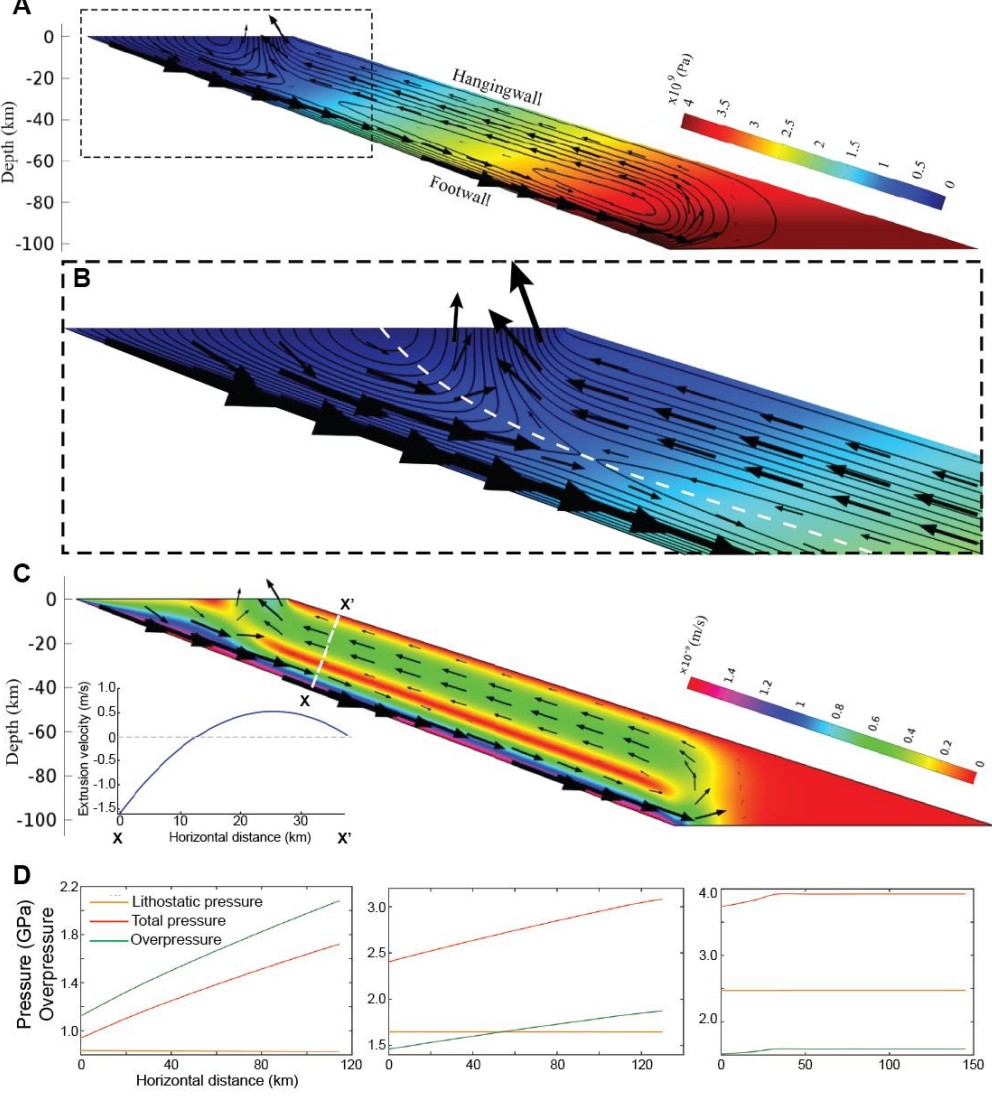


*Figure 2. Pressure and velocity maps and graphs for a UTC with α=20º, $W_m$=100 km, $W_b$=150*
*km, $U$ = 4 cm/yr, and η = $10^{21}$ Pa s. A – Velocity vectors and streamlines superimposed on*
*pressure map (background colour and colour bar), where two distinct flow circuits can be*
*recognized, one above and the other below -30 km. Also note asymmetry of flow relative to*
*channel, with upward return flow concentrated nearest the hanging wall. B – Zoom of the*
*topmost domain of the channel (marked by dashed rectangle in A). Note the convergence toward*
*the surface between a shallow flow (mostly on the footwall side and carrying lower pressure and*
*overpressure as seen in D) and a deep flow (mostly on the hanging wall side and carrying higher*



*pressure and overpressure as seen in D). White dashed line separates downward and upward flows. C – Velocity vectors superimposed on velocity coloured map (colour bar for scale). Note the red stripe of lower velocity closer to the footwall, which corresponds to the line of flow reversal in the model. Inset in C showing a velocity profile across the channel (marked by white dashed line and X-X'). D – graphs showing P, PL and TOP = P/PL at -30, -60 and -90 km. Note that the highest overpressure occurs at the shallowest depth, and increases toward the hanging wall (except at -90 km).*

Varying $W_m$ with other parameters constant and $\eta = 10^{21}$ Pa s shows that the UTC develops overpressure in the entire range of $W_m/W_b = W_m^* = 25/150$ to $100/150$ km (Fig. 3A). *TOP* is inversely proportional to $W_m^*$, and can be as high as 10 for $W_m^* = 0.17$ at depths between 20 and 60 km, with the highest *TOP* at 20 km depth.

*TOP* is sensitive to $\alpha$ in a UTC under a given set of values for $W_m$, $\eta$ and $U$ (Fig. 3B). The results plotted in Fig. 3B show *TOP* > 1 for $15^\circ < \alpha \leq 30^\circ$. *TOP* is maximal at $\alpha = 20\text{-}25^\circ$, reaching 1.7 at depths between 40 and 60 km.

The plot in Fig. 3C shows increase in *TOP* with increase in $U$, from *TOP* $\approx$ 1.5 at $U = 2\text{-}5$ cm/yr (current Indian velocity), to *TOP* $\approx$ 11 when $U = 20$ cm/yr (Indian velocity at 60-70 Ma).

The simulations show a near-exponential variation of *TOP* with $\eta$ (Fig. 3D), which we use to constrain the viscosity in the Himalayan collision zone.

Above we presented numerical simulations for $\eta = 10^{21}$ Pa s, typically applicable to the Himalayan tectonic setting. However, we ran additional simulations with different viscosities, and a set of results is presented for a viscosity of $10^{22}$ Pa s (Fig. 4). $\eta = 10^{22}$ Pa s induces much higher overpressure, especially when the mouth width decreases, and when the underthrusting velocity increases to velocities that have been estimated to exist at 60-70 Ma.

Taken together, the results shown in Figs. 3 and 4 place constraints on the factors affecting overpressure. Extremely high values of *TOP* are obtained for $\eta > 10^{21}$ Pa s, $U > 5$ cm/yr, and $W_m^* < 0.50$.





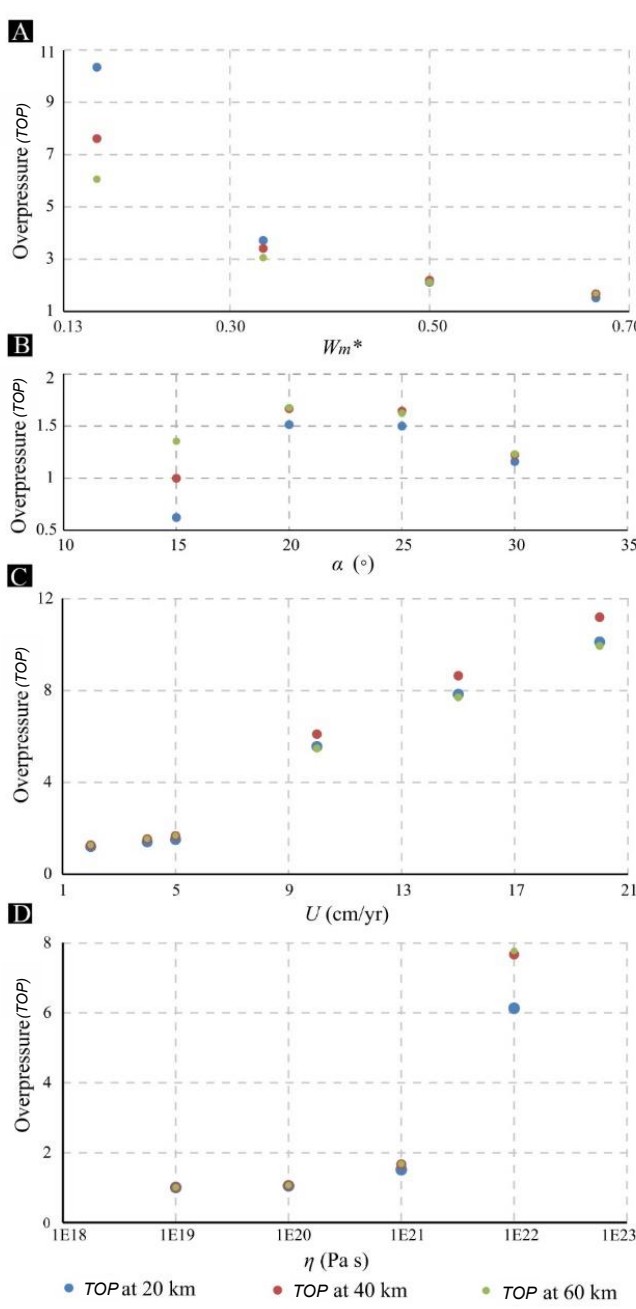

*Figure 3. Graphs showing the dependence of overpressure factor (TOP) on normalized width of channel mouth $W_m*$ (A), channel dip $\alpha$ (B), underthrusting velocity $U$ (C), and viscosity in the channel $\eta$ (D). For each tested variable, other values are kept constant: $W_m* = 100/150 = 0.67$ (except in A), $\alpha = 20^o$ (except in B), $U = 5$ cm/a (except in C), $\eta = 10^{21}$ Pa s (except in D).*

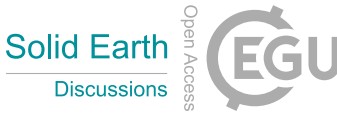


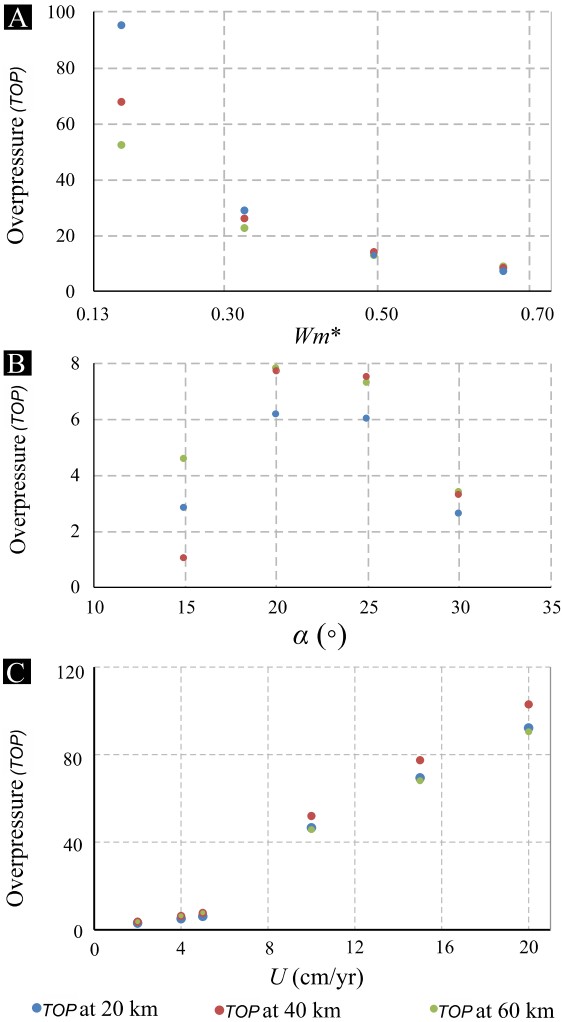

*Figure 4. Graphs showing overpressure factor TOP as a function of normalized channel's mouth width $W_m^*$ (A), channel dip $\alpha$ (B), and convergence velocity $U$ (C), for a viscosity $\eta = 10^{22}$ Pa s. For each tested variable, other values are kept constant: $W_m^* = 100/150 = 0.67$ (except in A), $\alpha = 20°$ (except in B), $U = 5$ cm/a (except in C). Comparison with Fig. 3 shows that $\eta = 10^{22}$ Pa s induces much higher overpressure, especially at smaller $W_m^*$ and higher $U$.*


Varying channel dip ($\alpha$) involves significant changes in the flow pattern, as shown in Fig.
5. For $\alpha = 15°$, the channel is dominated by downward flow, setting in a large-scale vortex in the
deeper level, and does not show conspicuous zones of ductile extrusion, which only occurs when
$\alpha > 15°$.



Figure 5. Simulations showing the effects of channel dip (α) on flow pattern.

Besides the results obtained for a channel base width of 150 km, and variable mouth

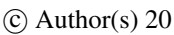


width, we also evaluated the effects of the channel base width on flow patterns and pressure
distribution, by running a set of numerical simulations with a base width of 200 km. The channel
flow shows similar patterns in the two cases, and small variations in pressure.

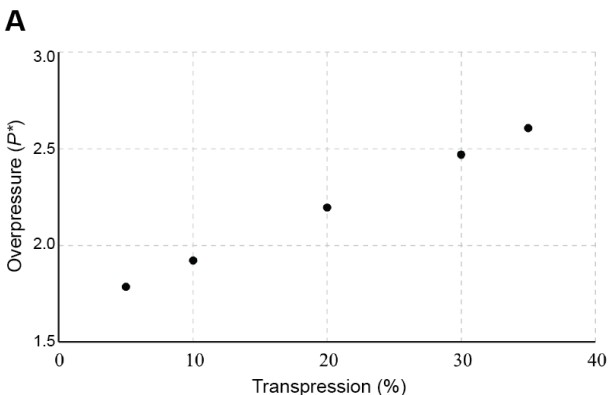


*Figure 6. Graphs showing the linear dependence of overpressure (TOP)(A) and extrusion*
*velocity (B) on transpression.*

*3.3. Effects of transpression on overpressure and flow*

We ran a set of simulations to investigate how much a transpressional movement across

the viscous channel might influence the magnitude of tectonic overpressure and, especially,
velocity at the channels mouth (extrusion velocity). Transpression in the numerical models was
introduced by setting the magnitude of horizontal velocity in excess of that corresponding to the




underthrusting movement. Fig. 6 shows a plot of *TOP* as a function of transpression, represented
as the ratio between horizontal velocity and non-transpressional horizontal component (ca.
1.49E-9 m/s). The numerical results indicate that: (1) transpression has appreciable effects on
overpressure, especially if transpression is large (> 20%); (2) transpression has great effects on
extrusion velocity, as shown in Figs. 6 and 7.

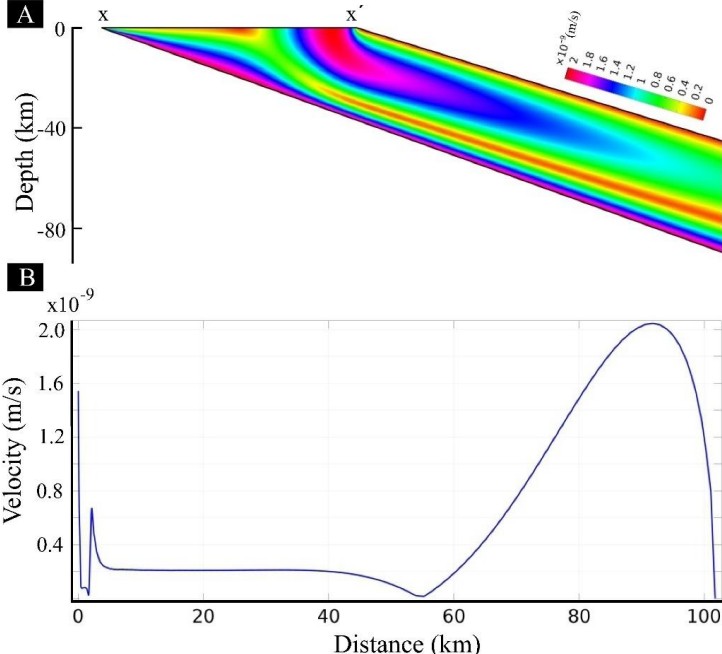


*Figure 7. A – Velocity map of a channel under transpression. X-X' marks the line along which*
*extrusion velocity was measured and plotted in B.*

*3.4. Viscous deformable walls*

We used a similar modelling approach to evaluate the magnitude of overpressure in

subduction channels confined by deformable walls, a model condition that closely replicates the
actual mechanical setting in the Himalayas. This model allows for both channel walls to deform
viscously, thus raising the question of how much overpressure they can retain inside the channel.
We developed the deformable wall models with a channel geometry similar to that in rigid wall



models, as shown in Figure 8A. The footwall and the hanging wall of the channel were
rheologically modelled as a viscous material, which provides a good approximation for
simulation of long term (millions of years) rheology of the lithosphere. Several earlier workers
have used viscous rheology to model continental scale deformation during India-Tibet collision.
The assumed viscosity values of the cold Indian craton range from $10^{23}$ to $10^{25}$ Pa s (e.g.
Jiménez-Munt and Platt, 2006; Yang and Liu, 2013), whereas that of Himalayan subducted
material ranges between $10^{20}$ and $10^{21}$ Pa s (e.g. Liu and Yang, 2003; Copley and Mckenzie,
2007). The viscosity ratio (viscosity walls/viscosity channel) is therefore in the order of $10^2$ to
$10^5$. In our modelling we chose a conservative value of the viscosity ratio equal to $10^3$, where the
walls and channel viscosities are $10^{23}$ and $10^{20}$ Pa s, respectively. We constrained the model
boundaries with kinematic conditions as in the reference model with rigid walls. The lateral and
the top boundaries of the footwall were subjected to a velocity of 4 cm/yr sub-parallel to the
channel, whereas the lateral vertical boundaries of the hanging wall were fixed with zero
horizontal velocity components, leaving the vertical component unconstrained. Its top boundary
was also left unconstrained, allowing the material to extrude upward freely. The wall-channel
interfaces had a no-slip condition.

Model results show channel flow patterns quite similar to those observed in rigid wall

models. The extrusion occurs along a region close to the hanging wall in the form of a Poiseuille
flow (Fig. 8A). It is noteworthy that the footwall undergoes little or no deformation, although
being deformable. The entire footwall underthrusts by translational motion parallel to the
channel. We calculated both the dynamic and the static pressures along the channel axis, and
plotted them as a function of depth (Fig. 8B). Similarly to rigid wall models, the dynamic
pressure here exceeds the static pressure by nearly 1.5 GPa. For example, the static pressure at a
depth of 60 km is about 1.5 GPa, whereas the corresponding dynamic pressure reaches 3 GPa.
The pressure plots clearly suggest that subduction channels with deformable walls can also give



rise to large tectonic overpressures. For a viscosity ratio of $10^3$, the deformable wall models are
found to be mechanically identical to rigid wall models.

We also used a lower viscosity contrast of $10^2$, and found that even at this relatively low

contrast there is significant overpressure in the subduction channel.

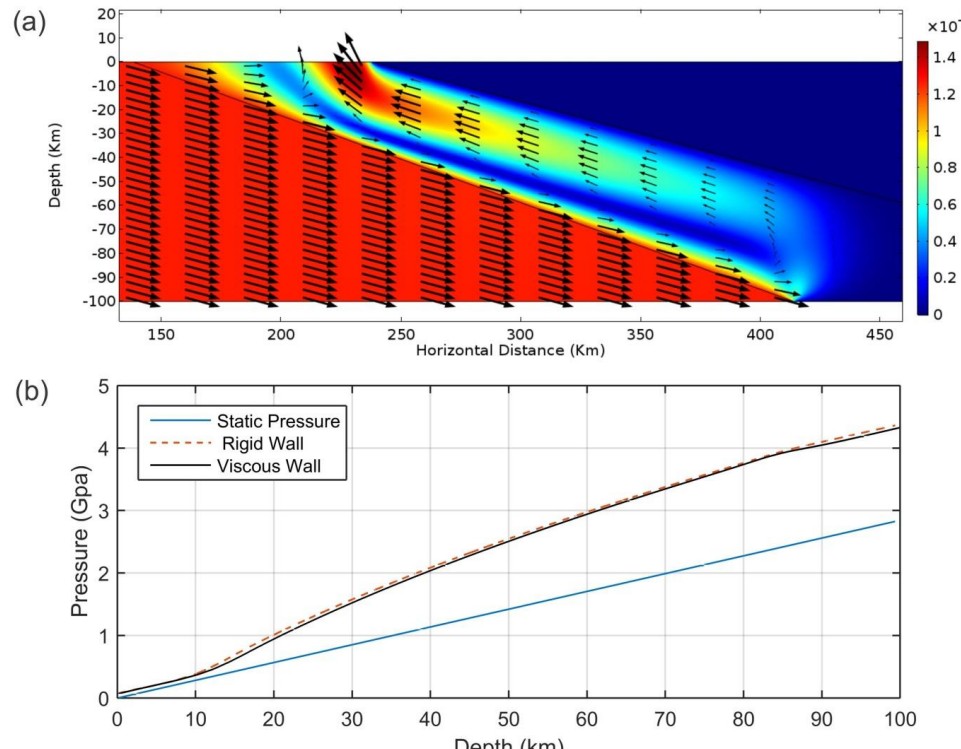

*Figure 8. A – Crustal flow patterns in viscous subduction channel and its deformable walls with*
*a viscosity ratio of $10^3$ (details of model boundary conditions in the text). B – Calculated plots of*
*pressure as a function of depth along the channel axis. Note that the dynamic pressure obtained*
*from deformable wall models with viscosity contrast 1000 closely follows that for channels with*
*rigid walls.*


*3.5. Condition at the bottom boundary*

This is a critical boundary condition because it is directly related to the retention of

overpressure. When we assign an outlet pressure (calculated lithostatic pressure at the depth of
the bottom wall) to the bottom wall, *TOP* does not develop in the whole channel.





### 4. Discussion


Given our incomplete knowledge of natural prototypes and the limitations of modelling

very complex systems, we have to distinguish between the theoretically and naturally possible
values of overpressure. The study here reported for a UTC shows that relevant parameters like
channel mouth width ($W_m$*), subduction dip ($\alpha$), underthrusting velocity ($U$) and viscosity ($\eta$)
can produce very high overpressure; however, these theoretically possible values have to be
constrained by the current knowledge of the Himalayas, in particular exhumation velocities and
spatial distribution, occurrence of HP and UHP rocks, and strength of the lithosphere bounding
the subduction channel. Despite the natural constraints imposed by our knowledge of the current
Himalayas, one cannot ignore that, under specific boundary conditions, geometrical
configurations and parameter sets that could have existed in the past (e.g. much higher
subduction velocity), high values of overpressure are theoretically possible, which should guide
us in the search of new evidence in the natural prototype.

Previous models have used two of the three main possible configurations of a subduction

channel: parallel-sided and downward tapering, which have been shown to produce relatively
small overpressure (*TOP* < 3) (e.g. Li et al., 2010). Here we investigated a different channel
geometry, the upward tapering channel. In fact, the parallel-sided geometry corresponds to $W_m$*
= 1, and can thus be considered an end-member of the UTC. Therefore, we can compare
numerical results of overpressure obtained for parallel-sided and UTC channels, by looking at
the graph in Fig. 3A where we vary $W_m$*. Our best explanation for this effect is that the narrower
the mouth the higher the flow confinement, which results in increased velocity gradient in the
channel flow, and therefore the dynamic pressure.

Previous models can explain channel flow, but neither account for the exhumation of HP

rocks (Rubatto et al., 2013), nor the exhumation velocities (Grujic et al., 2011) reported from the
Himalayas. Our UTC model provides an alternative explanation for the pressure required for



eclogite metamorphism (Hetényi et al., 2007; Zhang et al., 2014), and the process of rapid
exhumation. For exhumation by extrusion to occur in the subduction channel, the flow pattern
inside the channel must have a specific configuration, as in the UTC. In such a velocity
configuration, underthrusting and exhumation on the channel's footwall add to produce
enhanced overthrusting on the MCT, and above the MCT along the line of flow reversal.
Conversely, exhumation (upward flow) on the hanging wall is greater than underthrusting and
produces relative normal fault displacement on the STD, not because the block to the N of the
STD (hanging wall) moves down, but because the rocks south of the STD (footwall) move up
due to exhumation by extrusion.

Previous channel flow models can explain the exhumation mechanism, however they

leave a number of problems unaddressed. Here we raise some of these issues, pointing to our
UTC model as a unifying model to explain the GHS evolution:
(1) The classical channel flow model assumes that the entire GHS crustal mass thrusts up along

the MCT, with concomitant normal motion on the STD (Poiseuille flow). However, recent

studies have shown large-scale thrusts within the GHS (Grujic et al., 2011; Larson et al.,

2015), suggesting a more complex kinematics of the extrusion process. The UTC model we

propose here shows flow partitioning in the channel, leading to thrust-type shear localization

within the model GHS.

(2) A typical channel flow model fails to explain the occurrence of HP rocks (> 1.5 GPa) close to

the STD. Our UTC model yields an asymmetrical flow pattern in which HP or UHP

materials extrude along a narrow zone located close to the STD.

(3) The assumption of lithostatic pressure raises two main problems: (i) a conceptual problem,

because the subduction channel is dynamic, therefore the lithostatic and dynamic pressures

are not identical; and (ii) a practical problem, because the exhumation velocities are

calculated on the basis of depth estimated from $\rho g z$ (where $z$ is depth), and not normalized



387 by the overpressure. For instance, conversion of 2 GPa to depth using a static assumption

388 ($\rho gz$) yields a depth of ca. 70 km for a rock density of 2900 kg/m$^3$. However, the UTC flow

389 develops an overpressure in the order of 2 at much smaller depths, and thereby yields lower

390 exhumation rates, as compared to those calculated from petrologic modelling. Estimated

391 metamorphic paths should reflect the shape of the isotherms in the subduction channel,

392 which must have a relationship with velocity in order to carry cold rocks to depth, and

393 preserve the HP and UHP mineral parageneses during exhumation.

394 (4) Model velocities in the channel and at the channel's mouth must be consistent with the values

395 reported in the literature. Assuming lithostatic pressure, an exhumation rate of ~ 15 mm/yr

396 to a depth of at least 15 km was estimated by Ganguly et al. (2000). An estimate of 22–44

397 mm/yr, and increasing linearly with depth was provided by Grujic et al. (2011). According

398 to the UTC dynamic model, the assumption of lithostatic pressure where $TOP = 2$ yields an

399 overestimation of the exhumation velocity by a factor of 2. If this is the case, then the

400 velocity estimates have to be divided by two (15/2 = 7.5 mm/yr, and 33/2 = 16.5 mm/yr).

401 Our UTC model shows a high velocity layer with the materials flowing upward at a rate of

402 16 mm/yr at a depth of ca. 40 km, which is thus in agreement with the estimated average

403 exhumation. The velocity map in Fig. 6 reveals variations of exhumation rates with depth, as

404 predicted for the GHS in the Sikkim Himalaya by Ganguly et al. (2000), who showed that

405 the exhumation was rapid (15 mm/yr) to a depth of 15 km, and then decreased to ca. 2

406 mm/yr until a depth of 5 km. These values estimated for exhumation in the GHS constrain

407 the theoretical values of overpressure numerically obtained by varying the amount of

408 transpression. Transpression values > 10% imply velocities at the mouth (exhumation) much

409 larger than estimated for the GHS, therefore we conclude that transpression must be very

410 limited (< 10%).

411 (5) A critical issue regarding overpressure in a subduction channel is the strength of the channel



walls to support high overpressure values. The most debatable point in our modelling is the

use of rigid walls. For this discussion, we can compare the subduction channel to a pressure

vessel, in which the resistance of the vessel to internal pressure depends on two main

parameters: the strength of the vessel (the lithosphere hosting the subduction zone), and the

thickness of the pressure vessel walls (hoop stress). In nature, if the walls of the pressure

vessel (subducting and overlying lithospheres) are old and cold, which is the case in the

Himalayan collision, then their mechanical strength can be very high. If, additionally, the

cold and strong lithosphere is thick, then the walls of the subduction channel can support

high overpressure, as indicated by the numerical results with viscous deformable walls.

Given that the Indian plate and the TSS above the STD are almost undeformed (attesting to

the rigidity contrast between foot and hanging walls of the GHS) and thick, the channel

walls were assumed undeformable in the reference simulations. In order to investigate the

effects of viscous deformable walls on tectonic overpressure, we used viscosity contrasts

(viscosity of channel walls/viscosity of subduction channel) down to 100, which are well

within the accepted values of lithosphere viscosity (up to $10^{23}$ Pa s) and subducted material

(down to $10^{19}$ Pa s). These simulations indicate that viscosity contrasts of 1000 or 100 do not

change significantly the overpressure obtained with rigid walls. Another critical issue in

overpressure build-up is the condition at the bottom boundary: if an outlet pressure is

assigned to the bottom wall, then this boundary behaves as a leaking boundary that cannot

retain dynamic pressure. However, the cold, thick and strong lithospheres that comprise the

Indian and Eurasian plates are a good argument against a leaking bottom boundary in a flat-

ramp geometry such as the Himalayan collision zone.

(6)  In order to explain the non-linear variation of overpressure with channel dip ($\alpha$) we need to

analyse the variations of channel flow patterns with increasing $\alpha$ (Fig. 5). For low $\alpha$ values

(15º), the underthrusting motion drags materials to a larger extent into the downward flow,



and produces a large vortex in the deeper channel, where the curl dominates the flow field.
Consequently, the dynamic pressure remains low. Note that flow divergence increases the
dynamic pressure. With increasing $\alpha$ (20º) the flow pattern is characterized by the
development of an extrusion channel on the hanging wall side, along which the material
extrudes upward with flow convergence at the mouth. Such a negative divergence in the
flow builds overpressure on the hanging wall side (Fig. 2D). With further increase in $\alpha$ the
extrusion channel widens, and causes the overpressure to drop, as it happens in a pipe flow.
This is the reason why the overpressure has a maximum at $\alpha$ around 20-25º.
(7) Inverted metamorphic grade has not been explained by previous models, but the UTC can
provide an explanation if one considers the flow pattern shown in Fig. 2B. HP and UHP
rocks can be exhumed by two flow cells, both inverting metamorphism because low-grade
rocks go down close to the footwall, and high-grade rocks are exhumed close to the hanging
wall.

We analysed the consistency between the numerical results and geological/geophysical
data to constrain the most probable viscosity and pressure, at the same time satisfying a
reasonable velocity at the channel's mouth (i.e. exhumation rates) (Fig. 7). On the one hand, the
viscosity of rocks comprising the lithosphere can vary between $10^{19}$ and $10^{23}$ Pa s. On the other
hand, overpressure is sensitive to the viscosity within the UTC, increasing exponentially with
increase in viscosity. Additionally, from the values shown in Figs. 3 and 4, the formation of HP
rocks can occur at very shallow levels if $\eta = 10^{21}$ Pa s. However, despite the relatively wide
range of possible viscosity values, $\eta > 10^{21}$ Pa s in a Himalayan UTC yields overpressures > 8.
This means that, for $\eta = 10^{22}$ Pa s, a rock metamorphosed at 50 km depth would record a total
pressure equivalent to the lithostatic pressure at a depth of 400 km, which is not acceptable on
the basis of our current knowledge of subduction zone dynamics. Therefore, we propose that the





viscosity in the subduction channel is probably in the range $10^{20} \le \eta \le 10^{21}$ Pa s.

Regarding the discrepancy between previous estimates of possible values of overpressure

and ours, we call attention to two factors: (1) we use a subduction channel geometry, the UTC,
not investigated previously; and (2) the values reported here are very large only for small $W_m{}^*$,
or $U > 5$ cm/a, or $\eta > 10^{21}$ Pa s. In other words, for relatively small tapering ($W_m{}^*$), average plate
tectonics velocities, and reasonable viscosities, the numerical results reported here for
overpressure are not excessive, but nevertheless still very important as a factor for depth
overestimation. The values used for the controlling parameters, $W_m{}^*$, $\alpha$ and $\eta$ are conservative;
in fact, the model channel in Fig. 1C shows rather small tapering as compared with the cross-
section in Fig. 1B, but, nevertheless, the model overpressure is still quite high, especially at low
depth.

The UTC simulations show that there is no need for gravitational collapse, buoyancy-

controlled crustal exhumation, or orogen-perpendicular pressure gradient induced by a
topographic gradient to explain simultaneous reverse and normal fault kinematics on MCT and
STD, or inverse metamorphic grade, or exhumation of HP rocks. We conclude that flow in a
UTC, without the need for topography or density contrasts, can be responsible for these three
simultaneous and seemingly paradoxical processes in the Himalayas.

The formation and exhumation of high (HP) and ultra-high (UHP) pressure rocks is a

persisting fundamental problem, especially regarding UHP rocks. The problem is even greater if
one assumes that pressure estimated from paleopiezometry can be converted directly to depth,
because then the UHP rocks must be exhumed from great depths. Several models have been
proposed for the exhumation of HP and UHP rocks in several orogens (e.g. Hacker and Gerya,
2013; Warren, 2013; Burov et al., 2014a, 2014b): channel flow (e.g. England and Holland, 1979;
Mancktelow, 1995; Grujic et al., 1996; Beaumont et al., 2001, 2009; Burov et al., 2001;
Raimbourg et al., 2007; Gerya et al., 2008; Warren et al., 2008; Li and Gerya, 2009); eduction



(e.g. Andersen et al., 1991; Kylander-Clark et al., 2012); buoyancy-driven crustal delamination
and stacking (e.g. Chemenda et al., 1995, 1996; Sizova et al., 2012); microplate rotation (e.g.
Hacker et al., 2000; Webb et al., 2008); trans-mantle diapirism (e.g. Stöckhert and Gerya, 2005;
Little et al., 2011; Gordon et al., 2012); and slab rollback (e.g. Brun and Faccenna, 2008;
Faccenda et al., 2009; Vogt and Gerya, 2014; Malusà et al., 2015). No model has so far provided
a complete and unique explanation. The UTC model presented here is a potentially unifying
model, because it shows that it is possible to form rocks recording HP or UHP at depths < 60 km
and to exhume them to the surface as a consequence of the flow configuration in the UTC.

**5. Conclusion**

The UTC model integrates and provides a robust physical explanation for a number of

landmark features in the Greater Himalayan geodynamics, such as simultaneous reverse and
normal faulting (channel flow), inversion of the metamorphic grade in the GHS, and exhumation
of HP/UHP rocks along a narrow conduit close to the STD. Viscous flow in a UTC involves
dynamic pressures in excess of lithostatic pressure, resulting in significant overpressure by a
factor more than 1.5, even at depths as shallow as 40 km. The UTC model predicts high pressure
(>1.5. GPa) metamorphism of underthrusted rocks, e.g. eclogitization, to occur above 60 km
depth. The UTC model shows that the GHS is segmented broadly into two sub-terrains with
contrasting pressures: wide southern and narrow northern terranes, with pressures less and
greater than 1.5 GPa, respectively. It further shows that temporal variations in channel dip may
promote ($\alpha > 15°$) or inhibit ($\alpha < 15°$) exhumation. Overpressure increases with increase in $U$,
from $TOP \approx 1.5$ for $U$ = 2-5 cm/yr (current Indian velocity), to $TOP \approx 11$ when $U$ = 20 cm/yr
(Indian velocity at 60-70 Ma), which means that in the past all the dynamic processes discussed
here may have been enhanced. We tested different model setups (e.g. parallel walls) and
boundary conditions (e.g. slip or no-slip condition at bounding walls), but these do not reproduce



the prototype. The UTC model shows that tectonic pressure alone can drive the extrusion of HP
rocks by channel flow. Viscous deformable walls do not affect overpressure significantly for
viscosity contrasts (viscosity walls/viscosity channel) in the order of 1000 or 100. If, during the
subduction process, the mouth width, or the dip, or the velocity, or the viscosity, or the
conditions at the boundaries change in space and time, then *TOP* will change accordingly, and
the exhumation mechanism (flow in the channel) and exhumation depth will also change.
*TOP* in a UTC is only possible if the condition at the bottom boundary is not outlet
pressure; otherwise it behaves as a leaking boundary that cannot retain dynamic pressure.
However, the cold, thick and strong lithospheres that comprise the Indian and Eurasian plates are
a good argument against a leaking bottom boundary in a flat-ramp geometry, which means that
overpressure can build up to high values in the GHS. The argument does not apply if the channel
is "open" at the bottom, because overpressure cannot be retained. This could be the case in
subduction zones where there is no evidence for return flow and exhumation concomitant with
subduction.
The numerical results reported here show that, under specific boundary conditions,
geometrical configurations, and parameter sets, high values of overpressure are theoretically
possible, which should guide us in the search of new evidence in the natural prototype to prove
or disprove the natural existence of high overpressure.

**Acknowledgements**
FOM benefited from a sabbatical fellowship awarded by FCT Portugal
(SFRH/BSAB/1405/2014). NM acknowledges DST-SRB, India, for providing a J.C. Bose
Fellowship. SG acknowledges funding for doctoral research from the University Grants
Commission (UGC/275/Jr Fellow (Sc.)). GR thanks Carleton University for research support.



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




**Appendix - Methods**
*Boundary conditions and model setup*

The boundary conditions needed to complete the mathematical formulation for numerical

simulations were as follows: (1) slab-parallel velocity applied on the underthrusting wall,
consistent with the horizontal velocity of the Indian plate (5 cm/yr, DeMets et al., 2010); (2) slip
condition on (parallel to) the bottom boundary (Nábělek et al., 2009); (3) no slip condition on the
hanging wall; (4) outlet condition with 1 atm pressure at the channel's mouth; (5) gravity applied
to the whole channel (~9.8 m/s$^2$); (6) constant density of the material in the channel = 2800
kg/m$^3$ (no phase changes in the models), representing the association felsic (mostly) and mafic
granulites carrying the eclogite pods. Given that the Indian plate and the TSS above the STD are
almost undeformed, attesting to the rigidity contrast between foot and hanging walls of the GHS,
the channel walls were assumed undeformable in the simulations, except those testing the effects
of viscous walls. In order to investigate flow kinematics and dynamic pressure in the channel, we
varied the following parameters: (1) channel viscosity ($\eta$), (2) underthrusting velocity ($U$), (3)
channel dip ($\alpha$), (4) channel mouth's width ($W_m$), and (5) viscosity of channel walls. The
viscosity in the channel was varied between 10$^{19}$ and 10$^{22}$ Pa s to cover a broad spectrum of
crustal viscosities, as reported in the literature (Beaumont et al., 2001; England and Houseman,
1989; Copley et al., 2011). The current convergence rate between India and Eurasia has been
estimated in the order of 5 cm/yr, however, given the wide range of estimated velocities (Feldl
and Bilham, 2006; DeMets et al., 2010), we ran numerical simulations varying $U$ between 2 and
20 cm/yr (6.34E-10 to 6.34E-9 m/s in the model). Channel dip was varied between 15 and 30º,
which broadly covers the geometry of the GHS shown in different geological sections. We
assumed $W_m$ = 25 to 100 km, and $W_b$ (width at the channel's base) = 150 or 200 km, from which
we define $W_m^* = W_m/W_b$. We tested a viscosity contrast (viscosity of channel walls/viscosity in
the channel) of 1000 to investigate the effects of viscous deformable walls on overpressure.





Despite varying all these parameters, the prime focus of our investigation concerned the
simulations with $U$ = 5 cm/yr, $\alpha$ = 20°, $W_m$ = 100 km and $W_b$ = 150 km, as they represent the
most common and conservative values regarding published data. We then use the numerical
results to constrain the viscosity, pressure and velocity in the channel, consistent with current
geological data and estimates.
The metamorphic processes occur in response to the total isotropic stress, called *dynamic*
*pressure*, which is a sum of the tectonic (Stokes) and lithostatic pressures ($\rho g z$, where $\rho$ is
density, $g$ is gravitational acceleration, and $z$ is depth) (Figs. A1 and A2). The dynamic pressure
results from the viscous flow driven by tectonic stresses in the gravity field. Using the present
mechanical model, we evaluate the dynamic pressure to explain the occurrence of high-pressure
rocks in the GHS, as a consequence of dynamic pressure in excess of lithostatic pressure at a
given crustal depth. We define an overpressure factor (*TOP*) as the non-dimensional ratio
between dynamic and lithostatic pressures. For a better understanding of overpressure in a UTC,
we carried out a parametric study of *TOP* as a function of $\eta$, $W_m$, $\alpha$, $U$, and effective
convergence velocity (horizontal velocity component > $U$).

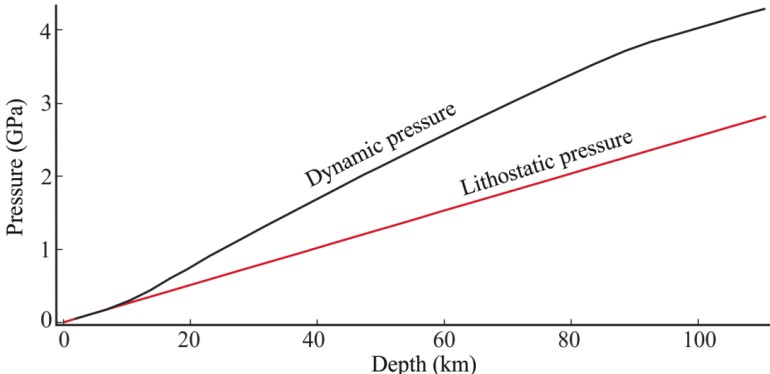


*Figure A1. Evolution of dynamic and lithostatic pressures in a UTC with $\eta = 10^{21}$ Pa s and $\rho =$*
*2800 kg/m³. The ratio dynamic pressure/lithostatic pressure corresponds to the overpressure*
*factor (TOP).*






*Figure A2. Overpressure in the UTC under the velocity field shown in Fig. 3.*


*Mathematical formulation*

The mathematical model used in the present work is based on the Navier-Stokes

equations for two-dimensional steady-state incompressible viscous flows:
$$\rho\left(\frac{\partial \mathbf{u}}{\partial t} + \mathbf{u} \cdot \nabla \mathbf{u}\right) = -\nabla p + \eta \nabla^2 \mathbf{u} + \mathbf{F} \qquad (1)$$

$$\nabla \cdot \mathbf{u} = 0 \qquad (2)$$

where $\mathbf{u}$ is the velocity vector, $p$ the pressure, $\rho$ the density, $\eta$ the dynamic viscosity and $\mathbf{F}$ the
external body force (gravity). $\rho$ and $\eta$ are constant. Then, defining the scaled variables $\bar{x} = x/L$,
$\bar{u} = u/U$, $\bar{p} = p/P$ and $\bar{t} = t/T$, in terms of the characteristic length $L$, velocity $U$, pressure $P$
and time $T = L/U$, Eqs. (1) and (2) become:
$$\frac{\partial \bar{\mathbf{u}}}{\partial \bar{t}} + \bar{\mathbf{u}} \cdot \nabla \bar{\mathbf{u}} = -\mathrm{Eu}\,\bar{\nabla}\bar{p} + \frac{1}{\mathrm{Re}}\,\bar{\nabla}^2 \bar{\mathbf{u}} \qquad (3)$$

$$\bar{\nabla} \cdot \bar{\mathbf{u}} = 0 \qquad (4)$$

where $\mathrm{Re} = \rho U L / \eta$ and $\mathrm{Eu} = P/\rho U^2$ are, respectively, the Reynolds and Euler numbers. For flows
at low characteristic velocity $U$ and high viscosity $\eta$, inertial terms Eu and Re in Eq. (3) become
negligible. We thus obtain the Stokes approximation of the momentum equation for quasi-static



(creeping) flows, which in dimensional form and under a gravity field reads:

$-\nabla p + \eta\,\nabla^2 u + \mathbf{F} = 0$ (5)

The Stokes equations were solved on the 2-D domain illustrated in Fig. 1C, which was

filled with an incompressible viscous linear fluid. The flow equations, with the boundary
conditions specified, were solved in the primitive variables $\mathbf{u}\equiv(u,v)$ and $p$ over a finite element
mesh, using the algorithm for incompressible Stokes flows implemented in COMSOL.