# Peer review of "Channel flow, tectonic overpressure, and exhumation of # high-pressure rocks in the Greater Himalayas"

_Solid Earth, 2018_

## Referee Comment (RC1) · Anonymous Referee #1 · 31 May 2018

In the manuscript by Marques et al., numerical models with a critical geometrical configuration, i.e. the upward-tapering channel, have been systematical studied, which show large tectonic overpressure in the confined subduction channel. In addition, series of parameters may regulate the flow kinematics and dynamic pressure inside the channel, including (1) channel viscosity, (2) underthrusting velocity, (3) channel dip, (4) channel mouth's width, and (5) rigid versus deformable channel walls.

The study aims to resolve the geodynamics of the GHS. Based on the numerical experiments, an upward-tapering channel (UTC) model is proposed to account for the combination of well-known geologic features, including simultaneous reverse and normal faulting, inversion of the metamorphic grade in the GHS, and exhumation of HP rocks along a narrow conduit close to the STD. In particular, this study focuses on the evaluation of dynamic overpressure to explain the formation and exhumation of high-pressure rocks in the GHS.

The paper is generally interesting with careful numerical investigations of the dynamic pressure evolution in such a model with 'special' geometry, which has not been studied before (as I know). In these aspects, I think it is worth publishing.

For this manuscript, my main concern is about the extremely high overpressure (even larger than ten times of the lithostatic pressure). I think such high values should be strongly related to the specific model geometry (upward taper), which is the most favorable condition for the overpressure. This geometry is constructed by comparing with the GHS geometry (cf. Figure 1b, c). In this model setup, the Tethys Himalaya (TSS shown in Figure 1b) is considered as a strong wall (rigid or rheologically strong); however, if TSS is weak, i.e. comparable to the GHS, then the channel geometry will be downward taper or parallel walls, similar to the general subduction channels. In the latter case, I do not think such high overpressure could be obtained.

Secondly, there are many previous numerical studies for the tectonic overpressure. I think a general discussion and comparison is required. In this manuscript, the authors just comment that 'previous models have used two of the three main possible configurations of a subduction channel: parallel-sided and downward tapering, which have been shown to produce relatively small overpressure (TOP < 3) (e.g. Li et al., 2010).' In my opinion, for the overpressure TOP = dynamic pressure / lithostatic pressure, the value 3 is quite large, which indicate the dynamic pressure is three times of the lithostatic. In this case, the rocks at 30km depth can obtain the pressure of up to 90 km. So it is better not to consider it as 'small overpressure'.

In order to avoid the possible misleading for the UHP community, I suggest adding a separate section at the end of the paper to discuss the specific conditions and/or
model limitations of the current more theoretical studies. Actually, many explanations have already been included in the main text (located in different sections).

The paper is generally well-written, without clear typos, etc.

---

## Author Comment (AC1) · 16 Jun 2018

Response to the Reviewer's comments

Reviewer#1 1. "For this manuscript, my main concern is about the extremely high overpressure (even larger than ten times of the lithostatic pressure). I think such high values should be strongly related to the specific model geometry (upward taper), which is the most favorable condition for the overpressure. This geometry is constructed by comparing with the GHS geometry (cf. Figure 1b, c). In this model setup, the Tethys Himalaya (TSS shown in Figure 1b) is considered as a strong wall (rigid or rheologically strong); however, if TSS is weak, i.e. comparable to the GHS, then the channel

geometry will be downward taper or parallel walls, similar to the general subduction channels. In the latter case, I do not think such high overpressure could be obtained." The Reviewer is right, the high values of TOP do not apply if the boundary conditions are different. We have now made clearer, in a separate section, the effects of boundary conditions.

2. "Secondly, there are many previous numerical studies for the tectonic overpressure. I think a general discussion and comparison is required. In this manuscript, the authors just comment that 'previous models have used two of the three main possible configurations of a subduction channel: parallel-sided and downward tapering, which have been shown to produce relatively small overpressure (TOP < 3) (e.g. Li et al., 2010)." General discussion and comparison added.

3. In my opinion, for the overpressure TOP = dynamic pressure / lithostatic pressure, the value 3 is quite large, which indicate the dynamic pressure is three times of the lithostatic. In this case, the rocks at 30km depth can obtain the pressure of up to 90 km. So it is better not to consider it as 'small overpressure'." Corrected.

4. "In order to avoid the possible misleading for the UHP community, I suggest adding a separate section at the end of the paper to discuss the specific conditions and/or model limitations of the current more theoretical studies. Actually, many explanations have already been included in the main text (located in different sections)." Separate section added to the Discussion.

The Discussion section has been fully revised following the Reviewer's comments.

Lisbon, 16 June 2018

Fernando Ornelas Marques on behalf of all authors

Please also note the supplement to this comment:
https://www.solid-earth-discuss.net/se-2018-37/se-2018-37-AC1-supplement.pdf

[Figure]

[Figure]

**Supplement:**

[revised manuscript text omitted]

**4. Discussion**

**4.1. Comparison with previous work**

The occurrence of TOP has received much attention in the geological literature (e.g. Rutland, 1965; Mancktelow, 1993, 1995, 2008 and references therein; Petrini and Podladchikov, 2000; Schmalholz and Podladchikov, 2013; Schmalholz et al., 2014b). TOP has been argued to exist in both hard (Mancktelow, 1993) and soft (Mancktelow, 1995) layers, and its occurrence has been predicted by force balance considerations independent of rheology (Schmalholz and Podladchikov, 2013; Schmalholz et al., 2014a, 2014b). We have previously explored (Marques et al., 2018) the occurrence of TOP in higher viscosity layers intercalated in lower viscosity layers (layer-parallel shortening of a rheologically stratified lithosphere), and in a lower viscosity layer between higher viscosity walls (subduction zone).

Previous work has investigated the occurrence of TOP at all scales: (1) local variations in pressure (e.g. Mancktelow, 1993; Tenczer et al., 2001; Taborda et al., 2004; Marques et al., 2005a, 2005b, 2005c, 2014; Schmid and Podladchikov, 2003, 2004; Ji and Wang, 2011; Schmalholz and Podladchikov, 2014; Tajčmanová et al., 2014, 2015; Angel et al., 2015), which in many cases is the natural consequence of the use of Stokes flow in the model, similarly to the numerical model used in the present study; (2) TOP in subduction zones (e.g. Li et al., 2010; Reuber et al., 2016). Given the great dependence of pressure on geometry, boundary and ambient conditions, and flow pattern, we cannot compare our Stokes flow models directly with the cited self-consistent geodynamic models, because in these the controlling parameters are combined with many other variables and parameters that act simultaneously and change with time. Therefore, we analysed, separately, the effects of the various parameters and boundary conditions on pressure in order to gain a better understanding of the effects of each of them.

TOP has been investigated as a function of the tectonic environment (e.g. Stüwe and Sandiford, 1994; Petrini and Podladchikov, 2000; Vrijmoed et al., 2009; Pleuger and

Podladchikov, 2014; Schmalholz et al., 2014a), and geometrical effects on TOP have also been addressed (e.g. Schmalholz and Podladchikov, 1999; Moulas et al., 2014), e.g. in downward tapering (e.g. Mancktelow, 1995, 2008 and references therein) and parallel-sided subduction channels, which have been argued to be the most appropriate configurations to model natural subduction zones. However, given the complexity and unsteady nature of subduction zones, the subduction channel can adopt all possible configurations, and the strictly parallel-sided configuration should be considered an exception rather than a rule, especially if we consider the

3-D, non-cylindrical, nature of subduction zones. Previous models have used two of the three main possible configurations of a subduction channel: parallel-sided and downward tapering, which have been shown to produce $TOP < 3$ (e.g. Li et al., 2010; Reuber et al., 2016). Here we investigated a different channel geometry, the upward tapering channel. In fact, the parallel-sided geometry corresponds to $W_m* = 1$, which can thus be considered an end-member of the UTC.

[revised manuscript text omitted]

An important question regarding TOP in nature still persists: why do we not see TOP in all subduction zones around the globe? On the one hand, our simulations indicate that roll-back subduction (transtension, in opposition to the favourable transpression) is unfavourable for the development of TOP. In contrast, collision-type subduction zones, like the Himalaya, with intervening old, cold and strong lithospheres are favourable for TOP. On the other hand, the recognition of TOP depends on methods and analytical technology, as shown by the most recent literature on petrology. There is growing evidence that TOP is recorded by minerals, as shown by Tajčmanová et al. (2014), Tajčmanová et al. (2015), Moulas et al. (2013, 2014) and Angel et al. (2015) (see also Menant et al., 2018). Constraints from host-inclusion elasticity show that TOP can greatly depart from lithostatic pressure; Angel et al. (2015) showed that deviations from lithostatic pressure in excess of 1 GPa can be readily produced in quartz inclusions within garnet in metamorphic rocks.

*4.3. Comparison between model and nature*

Inspired by the cross-section of the natural upward tapering channel shown in Fig. 1b, we investigated the effects of this geometry on TOP, and use it to find new explanations to the problems raised by the Himalayan geodynamics.

[revised manuscript text omitted]

---

## Referee Comment (RC2) · Anonymous Referee #2 · 21 Jun 2018

The study presented here investigates the applicability of the upward-tapering channel (UTC) model to understand the dynamics of the Greater Himalayan Sequence (GHS). In particular, the authors perform 2D numerical simulations of UTC to explain multiple pieces of evidence for the GHS, including inverted metamorphic grade in the channel, formation and exhumation of HP/UHP rocks close to the South Tibetan Detachment (STD), fault kinematics or tectonic overpressure.

The UTC simulates a flat-ramp geometry of the main underthrust faults (MCT and STD). Simulations predict that a UTC can account for high tectonic overpressure (>2), and exhumation of HP and UHP rocks along the channel's hanging wall. They also

constrain the GHS's viscosity to be less than 10ˆ21Pa.s

The authors also perform an insightful sensitivity analysis of the overpressure factor (TOP) to the parameters and factors varied, which include: the slab-parallel velocity, the channel viscosity, the geometry of the channel (mouth width, angle), and fixed or viscously deformable channel walls. Thus, they discuss the theoretical values of overpressure, and find that tectonic overpressure increases exponentially with decrease in UTC's mouth width, and with increase in underthrusting velocity and channel viscosity.

The manuscript has a well defined structure, with clear and well documented results and conclusions. I recommend the manuscript to be published in Solid Earth with minor modifications, and I provide below some comments that can be easily be addressed.

Some questions:

1. Have the authors tried running simulations with variable density walls in cases with deformable walls (i.e. channel density different from wall density)? How will that affect the pressure field?

2. What are the boundary conditions at the top of the channel (i.e. free surface/outflux)? I could not find this mentioned anywhere in the text.

Minor points:

Paragraph 124-130: the magnitude of the effective viscosity. What are the limitations of choosing an effective viscosity approach vs an Arrhenius approach? i.e. non-linear effect.

Line 152-153: 1-2 sentences with the main conclusion from Marques et al 2018 could help the readers here.

Line 164: Reference for "has been accepted as a simple but effective approximation. . ."

Line 168: References for COMSOL

[Figure]

Line 239: The authors can extend this section further by comparing the effect on TOP of all these factors (see Fig 3,4). Which one has the strongest effect? This can open a discussion on how these factors evolved/change in time in the Himalayan region. Viscosity?

Line 261: Yes, alpha affects the flow pattern, but not so much the TOP. Why is it that?

Line 279: Is it not clear how the transpression was calculated/set. A schematic/clearer sentence is needed.

Line 311: The hanging-wall-channel interface is no-slip (i.e. u=0). Because the footwall-channel interface has a prescribed U0. Or is that not the case?

Line 323: The TOP/pressure can be plotted in Figure 8B for evidence.

Section 3.5: Yes, an important condition, but the authors show no results. A figure with these simulations results can be added to the Appendix.

Line 351: Possible configurations: parallel-sided and downward tapering - what are the differences and the outcomes, limitations? Not all readers are familiar with the details of these models.

Paragraph 383-393: There is previous work to suggest that pressure-depth conversion in subducted rocks is not necessarily correct (i.e. Yamato and Brun, 2016). Should add some references.

Line 429: Experiments not shown.

Line 462: How does this estimate of viscosity in the channel relate to previous published estimates?

Paragraph 473-478: References for these hypotheses.

Paragraph 479-494: The authors list the proposed models for exhumation of HP and UHP rocks across various tectonic settings/orogens (with different geometries, rheology, boundary conditions). The UTC model can be applied to the Himalayas, but it's not clear that it can be considered a unified model in other settings. The authors should make that clearer.

Figure 1: C) Numerical model setup. Since boundary conditions (moving/fixed walls, bottom boundary conditions) are important in this study, it would be useful if they could be represented graphically in Figure 1C.

Figure 2: D) the plots are confusing because they do not have the same vertical axis.

Figure 3,4: Each simulation gave 3 values for TOP at different depths. What do these values represent? i.e. maximum/average TOP at the respective depth? same vertical axis would be desirable (currently apparent) and would allow comparison of factors on the overpressure. For example, which factor has the strongest effect on TOP? Green points are not distinct enough from blue markers.

Figure 5: What are the other reference parameters for these simulations? viscosity, Wm* etc.

Figure 6: missing panel B (possibly just a format/download issue)

Figure 7: How much transpression in this case? How was the transpression calculated from the velocity profile? Figure 8B: TOP should also be plotted in this case, and the case with viscosity contrast 1e2.

[Figure]

---

## Author Comment (AC2) · 29 Jun 2018

Response to the Reviewer#2's comments Comments in italic TNR, and responses in Arial blue

Main comments 1. "Have the authors tried running simulations with variable density walls in cases with deformable walls (i.e. channel density different from wall density)? How will that affect the pressure field?" The density is assumed constant throughout. Its variations would affect lithostatic pressure by a negligible amount (e.g. choosing = 2700 kg m-3 would decrease lithostatic pressure at a depth of 100 km by about 3%), and change by a small amount the dynamic pressure through the body force term in

the Stokes equation. Both effects are negligible in a first-order model such as ours. In the case of deformable walls, we parameterize their behaviour purely in terms of dynamic viscosity, and therefore a minor variation of density (through the change in lithostatic pressure) would be negligible.

2. "What are the boundary conditions at the top of the channel (i.e. free surface/outflux)?I could not find this mentioned anywhere in the text." The condition at the top of the channel is given in the Appendix, but we have now added it to the main text. We also added the condition at the bottom boundary. New text added: slip condition on (parallel to) the bottom boundary (Nábělek et al., 2009), and outlet condition with 1 atm pressure at the channel's mouth.

Minor comments Paragraph 124-130: the magnitude of the effective viscosity. What are the limitations of choosing an effective viscosity approach vs an Arrhenius approach? i.e. non-linear effect. The choice of the linear viscous rheology removes the effects of strain rate on the viscosity, which might affect the TOP. We can anticipate that at high velocities the viscosity would be lower, and so also the TOP.

Line 152-153: 1-2 sentences with the main conclusion from Marques et al 2018 could help the readers here. The new paragraph reads: This study builds on the conceptual work by Marques et al. (2018) on tectonic overpressure, in which the main conclusions are that TOP depends critically on boundary conditions (e.g. upward tapering channel can produce large TOP, whereas an outlet condition at the bottom prevents TOP from developing) and on critical parameters like strain rate and viscosity.

Line 164: Reference for "has been accepted as a simple but effective approximation" We have clarified the statement as follows: We have modelled the subduction channel ... with an incompressible linearly viscous fluid. The assumptions of incompressibility and linearity considerably simplify the model, and constitute standard procedure in many geophysical and geodynamic problems (cf, e.g. Ranalli, 1995; Turcotte and Schubert, 2014).
**Line 168: References for COMSOL Reference added: https://www.comsol.com/**

Line 239: The authors can extend this section further by comparing the effect on TOP of all these factors (see Fig 3,4). Which one has the strongest effect? This can open a discussion on how these factors evolved/change in time in the Himalayan region. Viscosity? The new paragraph reads: Given that the code is based on the Stokes' equation, viscosity and velocity play a fundamental role on the development of TOP. However, the the flow configuration is also critical, because velocity depends on the divergent of the velocity gradient in Stokes' equation. Furthermore, the flow configuration also depends critically on the boundary conditions, therefore some conditions favour the development of TOP (e.g. narrow channel mouth in an upward tapering channel) and others prevent it (e.g. an outlet condition at the bottom boundary).

Line 261: Yes, alpha affects the flow pattern, but not so much the TOP. Why is it that? We still do not have a sound explanation for this model result

Line 279: Is it not clear how the transpression was calculated/set. A schematic/clearer sentence is needed. The new paragraph reads: Transpression was set by adding an extra horizontal velocity component that made the velocity vector less steep than the moving subduction footwall

Line 311: The hanging-wall-channel interface is no-slip (i.e. u=0). Because the footwall-channel interface has a prescribed U0. Or is that not the case? The conditions at the hanging and foot walls are independent.

Line 323: The TOP/pressure can be plotted in Figure 8B for evidence. Now plotted. Sentence replaced with this new one: "The viscosity ratio (viscosity walls/viscosity channel) is therefore in the order of 102 to 105. In our modelling we chose a conservative value of the viscosity ratio equal to 102, where the walls and channel viscosities are 10E23 and 10E21 Pa s, respectively."

Section 3.5: Yes, an important condition, but the authors show no results. A figure
with these simulations results can be added to the Appendix. Figure added. Section now rewritten as follows: "3.5. Condition at the bottom boundary This is a critical boundary condition because it is directly related to the retention of overpressure. When we assign an outlet pressure (calculated lithostatic pressure at the depth of the bottom wall) to the bottom wall, TOP develops, but with lower magnitudes in the whole channel (supplementary figure)."

Line 351: Possible configurations: parallel-sided and downward tapering - what are the differences and the outcomes, limitations? Not all readers are familiar with the details of these models. The differences and outcomes are given in Marques et al. (2018). Reference added to the revised ms.

Paragraph 383-393: There is previous work to suggest that pressure-depth conversion in subducted rocks is not necessarily correct (i.e. Yamato and Brun, 2016). Should add some references. References added

Line 429: Experiments not shown. Now shown in the Appendix

Line 462: How does this estimate of viscosity in the channel relate to previous published estimates? New text added reads: ... viscosity in the subduction channel is probably in the range  $1020 \le \tilde{r}A_1 \le 1021$  Pa s, in agreement with the estimates for Himalayan subducted material (between 1020 and 1021 Pa s) by Liu and Yang (2003) and Copley and Mckenzie (2007).

Paragraph 473-478: References for these hypotheses. References added

Paragraph 479-494: The authors list the proposed models for exhumation of HP and UHP rocks across various tectonic settings/orogens (with different geometries, rheology, boundary conditions). The UTC model can be applied to the Himalayas, but it's not clear that it can be considered a unified model in other settings. The authors should make that clearer. We rephrased the text as follows: The UTC model presented here is a new potential model to explain the exhumation of HP and UHP rocks, ...

SED
Figure 1: C) Numerical model setup. Since boundary conditions (moving/fixed walls, bottom boundary conditions) are important in this study, it would be useful if they could be represented graphically in Figure 1C. They are now given in the main text to not overload the figure

Figure 2: D) the plots are confusing because they do not have the same vertical axis. The curves would not be individually visible if the vertical axis had the same scale. Please note that the curves in Fig.2D, where the maximum values go up to 2.2 GPa (left most panel) would merge with one another if the vertical axes had a scale of 4 GPa, as in the right most panel. This is the reason why the three panels were prepared with their vertical axis at different values.

Figure 3, 4: Each simulation gave 3 values for TOP at different depths. What do these values represent? i.e. maximum/average TOP at the respective depth? same vertical axis would be desirable (currently apparent) and would allow comparison of factors on the overpressure. For example, which factor has the strongest effect on TOP? Green points are not distinct enough from blue markers. The three values for TOP represent overpressure at three different depths at the middle of the channel, respectively 30, 60 and 90km for each experiment. For appropriate visualisation of the curves, it was necessary to use vertical axes with different scales. Previous light green colour is changed to deep green.

Figure 5: What are the other reference parameters for these simulations? viscosity, Wm\* etc. New caption now reads: "Figure 5. Simulations showing the effects of channel dip ( $\alpha$ ) on flow pattern, keeping Wm=100 km, Wb=150 km, U = 4 cm/yr, and viscosity = 1021 Pa s."

Figure 6: missing panel B (possibly just a format/download issue) Problem solved

Figure 7: How much transpression in this case? How was the transpression calculated from the velocity profile? The figure shows a transient analysis of the velocity field, i.e. instantaneous flow in the channel under a given transpression rate, which is measured
as the ratio between horizontal velocity and non-transpressional horizontal component (ca. 1.49E-9 m/s). Please note that this parameter is an alternative expression of the ratio between wall normal and wall parallel velocity components. "Fig. 6 shows a plot of TOP as a function of transpression, represented as the ratio between horizontal velocity and non-transpressional horizontal component (ca. 1.49E-9 m/s)."

Figure 8B: TOP should also be plotted in this case, and the case with viscosity contrast 1e2. To keep the clarity in the diagram, we plot only for the viscosity ratio 102.

Lisbon, 29th June 2018 Fernando Ornelas Marques on behalf of all co-authors

Please also note the supplement to this comment: https://www.solid-earth-discuss.net/se-2018-37/se-2018-37-AC2-supplement.pdf

---

## Author Comment (AC3) · 1 Jul 2018

Our response to comment on Section 3.5 is erroneous. Where you read:

Section 3.5: Yes, an important condition, but the authors show no results. A figure with these simulations results can be added to the Appendix. Figure added. Section now rewritten as follows: "3.5. Condition at the bottom boundary This is a critical boundary condition because it is directly related to the retention of overpressure. When we assign an outlet pressure (calculated lithostatic pressure at the depth of the bottom wall) to the bottom wall, TOP develops, but with lower magnitudes in the whole channel (supplementary figure)."

You should read:

Section 3.5: Yes, an important condition, but the authors show no results. A figure with these simulations results can be added to the Appendix. Figure added. Section now rewritten as follows: "3.5. Condition at the bottom boundary This is a critical boundary condition because it is directly related to the retention of overpressure. When we assign an outlet pressure (calculated lithostatic pressure at the depth of the bottom wall) to the bottom wall, TOP does not develop."